# Lipoprotein Subclass Profile after Progressive Energy Deficits Induced by Calorie Restriction or Exercise

**DOI:** 10.3390/nu10111814

**Published:** 2018-11-21

**Authors:** Yu Chung Chooi, Cherlyn Ding, Zhiling Chan, Jezebel Lo, John Choo, Benjamin T. K. Ding, Melvin K.-S. Leow, Faidon Magkos

**Affiliations:** 1Clinical Nutrition Research Centre (CNRC), Singapore Institute for Clinical Sciences (SICS), Agency for Science, Technology and Research (A*STAR) and National University Health System, Singapore 117599, Singapore; chooi_yu_chung@sics.a-star.edu.sg (Y.C.C.); cherlyn_ding@sics.a-star.edu.sg (C.D.); Chan_Zhiling@sbic.a-star.edu.sg (Z.C.); jezebellzq@gmail.com (J.L.); melvin_leow@sics.a-star.edu.sg (M.K.-S.L.); 2Singapore Institute for Clinical Sciences (SICS), Agency for Science, Technology and Research (A*STAR) and National University Health System, Singapore 117609, Singapore; jc29128@gmail.com (J.C.); benjamin.ding@mohh.com.sg (B.T.K.D.); 3Department of Endocrinology, Tan Tock Seng Hospital, Singapore 308433, Singapore; 4Cardiovascular and Metabolic Disorders Program, Duke-NUS Medical School, Singapore 169857, Singapore; 5Lee Kong Chian School of Medicine, Nanyang Technological University, Singapore 636921, Singapore; 6Department of Physiology, Yong Loo Lin School of Medicine, National University of Singapore (NUS), Singapore 117593, Singapore; 7Department of Nutrition, Exercise and Sports, Faculty of Science, University of Copenhagen, 1958 Frederiksberg C, Denmark

**Keywords:** lipoproteins, triglyceride, cholesterol, negative energy balance

## Abstract

Weight loss, induced by chronic energy deficit, improves the blood lipid profile. However, the effects of an acute negative energy balance and the comparative efficacy of diet and exercise are not well-established. We determined the effects of progressive, acute energy deficits (20% or 40% of daily energy requirements) induced by a single day of calorie restriction (*n* = 19) or aerobic exercise (*n* = 13) in healthy subjects (age: 26 ± 9 years; body mass index (BMI): 21.8 ± 2.9 kg/m^2^). Fasting plasma concentrations of very low-, intermediate-, low-, and high-density lipoprotein (VLDL, LDL, IDL, and HDL, respectively) particles and their subclasses were determined using nuclear magnetic resonance. Total plasma triglyceride and VLDL-triglyceride concentrations decreased after calorie restriction and exercise (all *p* ≤ 0.025); the pattern of change was linear with an increasing energy deficit (all *p* < 0.03), with no evidence of plateauing. The number of circulating large and medium VLDL particles decreased after diet and exercise (all *p* < 0.015), with no change in small VLDL particles. The concentrations of IDL, LDL, and HDL particles, their relative distributions, and the particle sizes were not altered. Our data indicate that an acute negative energy balance induced by calorie restriction and aerobic exercise reduces triglyceride concentrations in a dose-dependent manner, by decreasing circulating large and medium VLDL particles.

## 1. Introduction

Cardiovascular disease (CVD), especially ischemic heart disease and stroke, is the leading cause of mortality worldwide, accounting for approximately one-third of global deaths [1]. Disturbances in lipid metabolism leading to unfavorable changes in the plasma lipid profile—such as increases in triglyceride concentration and small, dense low-density lipoprotein (LDL) particles, increases in total and LDL-cholesterol concentrations, and decreases in high-density lipoprotein (HDL) cholesterol concentration—can augment the risk of CVD [2].

The primary management for CVD currently involves lowering LDL-cholesterol concentration with drugs [3]. However, the beneficial effect of cholesterol-lowering therapy by using statins is no greater than 30%, which indicates that other important risk factors need to be taken into account [4]. Lifestyle factors, such as diet and physical activity, are often overlooked but are key to reducing CVD risk [5,6]. Weight loss, which involves tipping the balance between energy intake and energy expenditure to achieve a state of negative energy balance, has been shown to improve the blood lipid profile (lower triglyceride, total and LDL-cholesterol concentrations), in conjunction with a reduction in total body fat, intra-abdominal fat, and ectopic fat accumulation [7,8,9,10]. Clearly, however, diet and exercise can affect lipid metabolism acutely, even after just a single day and in the absence of weight loss [11,12]. The degree of negative energy balance that needs to be achieved to improve lipid metabolism, as well as possible dose-response relationships between a negative energy balance and the plasma lipid profile, remain elusive. Also, little is known about the comparative efficacy of the same energy deficit achieved by restricting energy intake or increasing energy expenditure.

This study was therefore aimed at evaluating the lipoprotein subclass profile, by using nuclear magnetic resonance (NMR) spectroscopy, after a single day of progressive negative energy balance (20% and 40% of daily energy requirements), induced by calorie restriction (CR) or aerobic exercise (EX). Lipoproteins vary in particle size and number, and assessing the lipoprotein subclass profile offers a valuable insight into CVD risk, beyond that captured by traditional risk factors such as triglyceride and cholesterol levels. Increased CVD risk has been associated with increased numbers of small dense LDL [13,14,15], large very low-density lipoproteins (VLDL), and small HDL [16]. Therefore, characterizing the changes in lipoprotein subclass profile is necessary to better understand the effects of diet and exercise on lipid metabolism, and thereby help design effective lifestyle interventions targeted at preventing CVD.

## 2. Methods

### 2.1. Subjects

A total of 32 healthy subjects (22 women and 10 men), aged 21–59 years old, with a body mass index (BMI) between 18 and 30 kg/m^2^, participated in this study: 19 were enrolled in the CR group, and 13 were enrolled in the EX group. Initially, 15 subjects were recruited in each group, at which point we performed an interim analysis and decided to recruit an additional 5 subjects in the diet group, because the effect size turned out to be smaller than the one we had assumed to adequately power the study. As a result, a total of 15 subjects were recruited in the exercise group (2 dropped out and did not complete all the trials, hence 13 completers were analyzed), and a total of 20 subjects were recruited in the diet group (1 dropped out and did not complete all the trials, hence 19 completers were analyzed). The subjects had no prior history of glucose intolerance, thyroid disease, hypertension or dyslipidemia, and they all had non-diabetic fasting blood glucose and HbA1c concentrations at screening. Subjects who were taking medications known to affect metabolic function (including oral contraceptives and hormone replacement therapy), tobacco products and consuming alcohol regularly, had evidence of significant organ system dysfunction or disease, or recent weight loss or gain (≥ 5% over the past 6 months) were excluded from the study. Ethics approval was obtained from the Domain Specific Review Board of the National Healthcare Group in Singapore (protocol # 2016/00660, approved on 25 August 2016), and all the subjects provided their signed, written, informed consent prior to enrolment. 

### 2.2. Experimental Design

This was a cross-over study, with subjects in each group performing three experimental trials in random order. Following the successful screening and enrolment in the study, the subjects visited the laboratory to have their resting metabolic rate and body composition assessed. Thereafter, they visited the laboratory on three additional occasions, separated by 5–10 days, and underwent three experimental trials in random order. For the CR group, the three trials included one control trial (isocaloric diet, zero energy balance; CR-0) and two trials of progressively increasing negative energy balance induced by calorie restriction (20% and 40% reduction of daily energy needs for weight maintenance; CR-20 and CR-40); with respect to physical activity, all CR trials were performed under resting conditions. For the EX group, the three trials included one control trial (rest, zero energy balance; EX-0) and two trials of progressively increasing negative energy balance induced by aerobic exercise (20% and 40% reduction of daily energy needs for weight maintenance; EX-20 and EX-40); with respect to caloric intake, all the EX trials were performed under isocaloric conditions.

### 2.3. Body Composition and Resting Energy Expenditure 

The subjects visited the laboratory in the morning, after having fasted overnight, and having abstained from strenuous exercise, alcohol, and caffeine on the previous day. Fat mass and fat-free mass were determined using dual-energy X-ray absorptiometry (DXA) on a Discovery QDR Series DEXA scanner (Hologic, Bedford, MA, USA). Following 30 min of bed rest, they had their oxygen consumption and carbon dioxide production measured continuously for 30 min, while breathing under a ventilated hood, and their resting metabolic rate (RMR) was determined using indirect calorimetry (Quark RMR; COSMED, Rome, Italy). 

### 2.4. Blood Sampling 

For all three experimental visits, which were scheduled ~1 week apart, the subjects visited the laboratory in the morning (~8 am on day 1), after having fasted overnight. They were instructed to abstain from alcohol and caffeine consumption on the previous day (day 0), and from performing any strenuous exercise on the preceding 3 days (to avoid potential delayed metabolic effects of exercise). The subjects recorded all the food and drink they had consumed on the day before they came to the laboratory (i.e., day 0) for their first metabolic testing visit and were instructed to replicate the same diet on the day preceding the remaining visits. All the subjects confirmed that they followed the diet prescription during their subsequent admissions; some minor deviations were not deemed important. On all occasions, vital signs (temperature, heart rate, and blood pressure) were obtained after 30 min of bed rest, and before any testing began. During day 1, the subjects remained in the laboratory and ate and rested or exercised according to the study protocol and trial (Figure 1). They were discharged after eating dinner on day 1, and were instructed to refrain from all food and drink and physical activity until they returned to the laboratory the next morning (~8 am on day 2) after having fasted overnight, when fasting blood samples were obtained from a forearm vein.

### 2.5. Diet and Exercise Interventions

Daily energy requirements for weight maintenance (i.e., isocaloric diet) were estimated for each subject by multiplying resting energy expenditure (i.e., RMR, measured at screening) by a factor of 1.4 [17]. Two meals (breakfast, served at 9 am, and dinner, served at 7 pm) were provided during each trial; the total energy content of those meals was equivalent to 60% of the total calculated daily energy requirements for weight maintenance (i.e., 20% at breakfast and 40% at dinner). These two meals were identical in all three trials (breakfast: turkey breast sandwich with soya milk; dinner: teriyaki chicken rice with mixed vegetables and peaches). Two snacks (tuna sandwich with orange juice) were given in the morning and afternoon to provide the additional energy required for weight maintenance (i.e., 20% at each snack, for a total of 40%). Subjects in the CR group rested during the morning and afternoon of day 1 and consumed either an isocaloric diet (control, CR-0) or a progressively energy deficient diet by withholding the first (20% calorie restriction, CR-20) or both (40% calorie restriction, CR-40) snacks.

Subjects in the EX group consumed an isocaloric diet during day 1 and either rested or performed aerobic exercise (two bouts, performed at approximately the same times as the two resting periods; Figure 1) to expend an amount of calories equivalent to 20% (EX-20) or 40% (EX-40) of their daily energy requirements for weight maintenance. Exercise was performed on a cycloergometer (Wattbike Trainer, Woodway, Rhein, Germany) at submaximal intensity, and the duration varied accordingly to expend the targeted amount of calories. Resistance varied so that the heart rate was within 60–80% of the maximum heart rate (HRmax) and cadence was maintained at 60–80 rpm. Gross energy expenditure was calculated using a prediction equation utilizing age, sex, weight, and heart rate during exercise [18], and net energy expenditure was calculated by subtracting RMR for the duration of the bout. The three experimental trials in each group were performed in random order. All meals were prepared in our metabolic kitchen. Each meal or snack contained 55% of total energy as carbohydrate, 27% as fat, and 18% as protein, so the relative composition of the daily diet was the same in both groups and all trials. This design made it possible to induce a progressively increasing negative energy balance (equal to 20% and 40%) by restricting dietary intake (CR group) or by increasing energy expenditure (EX group), at approximately the same time of the day, while keeping the last meal and the duration of fasting before metabolic testing the same (Figure 1). 

### 2.6. Sample Analyses

Plasma concentrations of very low-density lipoprotein (VLDL), intermediate-density lipoprotein (IDL), low-density lipoprotein (LDL), and high-density lipoprotein (HDL) particles and subclasses were determined using nuclear magnetic resonance (NMR) spectroscopy on a Vantera Clinical Analyzer at LipoScience (LabCorp, Morrisville, NC, USA) [19,20]. Lipoprotein particle concentrations and sizes were calculated as previously described using the LP3 algorithm [21]. The concentrations of the following nine lipoprotein subclass categories were measured: large VLDL (including residual chylomicrons, > 60 nm), medium VLDL (42–60 nm), small VLDL (29–42 nm), IDL (23–29 nm), large LDL (20.5–23 nm), small LDL (18–20.5 nm), large HDL (9.4–14 nm), medium HDL (8.2–9.4 nm), and small HDL (7.3–8.2 nm). The average VLDL, LDL, and HDL particle sizes (diameter in nm) were computed as the sum of the diameter of each subclass multiplied by its relative mass percentage. The reproducibility of NMR determinations, expressed as the coefficient of variation (CV) of repeated measurements on the same samples, was 11% for total VLDL, 4% for total LDL, and 2% for total HDL particle concentrations, 4% for VLDL size, 1% for LDL size and HDL size, 14% for large VLDL, <14% for large and small LDL subclasses, and <17% for large, medium and small HDL subclasses. Higher CVs were obtained for medium VLDL (21%), small VLDL (28%), and IDL (45%) particles, due to their typically low concentrations in plasma. Plasma triglyceride, VLDL triglyceride, and HDL-cholesterol concentrations were determined using NMR; the NMR-derived values are highly correlated (r > 0.9) with the respective measurements from a conventional lipid analysis [21]. Furthermore, the NMR determinations of lipoprotein particle sizes correlate well with the size estimations derived from conventional methods, e.g., gradient gel electrophoresis [21]. 

### 2.7. Statistical Analysis

Prior to all the statistical analyses, the distributional properties of the outcome measures were evaluated for normality using the Shapiro-Wilks test. Parametric tests were used for normally distributed data, and non-parametric tests (based on ranks) were used for non-normally distributed data. The primary goal of our study was to determine the effects of progressively increasing energy deficit induced by calorie restriction or aerobic exercise on plasma lipoproteins. This was accomplished using an analysis of variance for repeated measures within each study group. Statistically significant models were followed by: (i) trend analysis to describe the pattern of change in the outcome with a progressively increasing negative energy balance (i.e., linear or quadratic); and (ii) simple contrasts to compare each level of energy deficit against the control trial. All these analyses were preceded by preliminary between-group comparisons using the Student’s unpaired *t* test (or the Mann-Whitney U test) and the χ^2^ test, which confirmed that there were no significant differences between the CR and the EX groups in key prognostic and demographic variables at baseline. Statistical significance was accepted at *p* ≤ 0.05. Results are shown as means with standard errors, unless otherwise noted. Statistical analysis was performed using SPSS version 23 (IBM SPSS, Chicago, IL, USA).

## 3. Results

### 3.1. Subject Characteristics

The two groups did not differ in any of their baseline characteristics, such as age (*p* = 0.89), sex (*p* = 0.41), BMI (*p* = 0.10), percent body fat (*p* = 0.24), fat mass (*p* = 0.98), and RMR (*p* = 0.38), even though the two groups were not specifically matched for these attributes *a priori* (Table 1). Most subjects (30/32) were Singaporeans of Chinese descent (the remaining two included one Vietnamese and one Indian).

### 3.2. Energy Deficits Induced by CR and EX

Subjects in the CR group consumed 1793 ± 96 kcal/day in the CR-0 trial, and progressively less in the CR-20 and CR-40 trials (1434 ± 76 and 1076 ± 57 kcal/day, respectively). This resulted in energy deficits of 359 ± 19 and 717 ± 38 kcal, corresponding to energy balances of -20±0 and -40±0 % in the CR-20 and CR-40 trials, respectively.

Subjects in the EX group consumed 1688 ± 66 kcal/day in all trials, did not exercise in the EX-0 trial, and exercised progressively more in the EX-20 and EX-40 trials (36 ± 2 and 70 ± 4 minutes, at 73 ± 2 and 76 ± 2 % of HRmax, respectively). This resulted in energy expenditures of 323±12 and 673 ± 28 kcal, corresponding to energy balances of -19.2 ± 0.6 % and -39.9 ± 0.5 % in the EX-20 and EX-40 trials, respectively. 

Others have shown that the respiratory exchange ratio in untrained subjects performing moderate-intensity exercise in the fasted and fed states is around 0.9 [22], indicating that the composition of the exercise-induced energy deficits was approximately 33% fat and 67% carbohydrate. Although this is different from the composition of the diet-induced energy deficits (55% carbohydrate, 27% fat, and 18% protein), we opted to match the macronutrient composition of the diets consumed in the two groups, as we felt this small difference in the composition of energy deficits was unlikely to confound the interpretation of our results.

### 3.3. Lipoprotein Subclass Profile

There were no significant differences between the groups in terms of baseline lipid and lipoprotein concentrations and lipoprotein subclass profile (Table 2). The total plasma triglyceride and VLDL-triglyceride concentrations decreased after calorie restriction (both *p* ≤ 0.025) and exercise (both *p* ≤ 0.001); the pattern of change was linear (all *p* < 0.03) with no evidence of plateauing (i.e., no significant quadratic component, Table 2). Exercise did not affect HDL-cholesterol concentration (*p* = 0.913), but calorie restriction caused a marginally significant decrease (*p* = 0.049); however, none of the pairwise contrasts against the control trial reached significance (Table 2). 

The reduction in triglyceride concentrations after CR and EX was accompanied by significant decreases in the number of circulating large and medium VLDL particles (all *p* < 0.015), with no change in small VLDL particles (*p* > 0.64, Table 2). This resulted in a shift in the percent distribution of VLDL towards smaller particles (Figure 2), but the average VLDL particle size did not change significantly (Table 2). The concentrations of IDL, LDL, and HDL particles (total and subclasses), their relative distributions, and the average LDL and HDL sizes were not significantly affected by 20% or 40% energy deficits induced by CR or EX (Table 2 and Figure 2). 

## 4. Discussion

In this study, we assessed the effect of progressive energy deficits induced by CR or EX on lipoprotein subclass profile. We found that the concentrations of total plasma triglyceride and VLDL triglyceride decreased after CR and EX to a similar extent and in a similar fashion, with greater energy deficits producing greater reductions. Previous studies evaluating either acute exercise [23,24] or acute calorie restriction [25], or both [26], reached similar conclusions, but possible dose-response relationships between diet- and exercise-induced energy deficits have never been investigated. We demonstrated that our acute calorie restriction diet with a standard macronutrient ratio (55% carbohydrate, 18% protein, and 27% fat) lowered total and VLDL triglyceride concentrations to a similar extent as modified-fat diets [27,28]. The effect of different types of diet therapies on lipid profile has been thoroughly studied [29,30] but the effect of acute energy restriction on lipoprotein subclass concentrations and sizes has yet to be established. We report here that acute diet-induced energy deficits lower circulating triglyceride concentrations because of reducing medium and large VLDL particles; acute calorie restriction does not affect the concentrations and sizes of LDL and HDL particles. 

Regarding exercise, several studies have observed that the improvements in lipid metabolism were related to the amount of physical activity, and not independently of the intensity or duration of exercise or the improvement in fitness [31,32], suggesting that the energy expenditure of exercise is key for the beneficial effects on lipid metabolism. Indeed, we documented a linear pattern of change in total and VLDL triglyceride concentrations with no evidence of plateauing between energy deficits from 20% to 40% of daily energy requirements for both CR and EX. Most triglyceride in post-absorptive plasma is carried in the core of VLDL [11], which explains the very similar pattern of change in these two parameters. Previous studies have reported that acute exercise-induced reductions in circulating triglyceride concentrations require a minimum threshold of exercise energy expenditure to manifest [33], and that the magnitude of the effect plateaus above another energy expenditure threshold [34]. This implies that our two energy deficit levels induced by diet and exercise fell somewhere in between these thresholds, since the effects we observed were linear. At or below the minimum energy deficit threshold, however, exercise may be more effective than diet in reducing VLDL triglyceride concentrations [35]. The mechanisms for the exercise-induced reductions in VLDL particle number and VLDL triglyceride concentrations likely involve a decrease in the production of VLDL particles from the liver and an increase in the clearance of VLDL triglyceride from the periphery [36]. Lipoprotein lipase hydrolyses VLDL triglyceride to fatty acids that are taken up by peripheral tissues for oxidation or storage, and exercise may induce skeletal muscle lipoprotein lipase [11]. On the contrary, chronic calorie restriction (leading to weight loss) lowers plasma triglyceride and VLDL triglyceride concentrations predominantly by reducing hepatic VLDL triglyceride secretion [11]. Unfortunately, we did not assess the underlying mechanisms for the changes we observed in lipid and lipoprotein profile in our study. Regardless, the reduction in circulating triglyceride after both diet- and exercise-induced energy deficits is in agreement with a previous report [26] and can be expected to reduce triglyceride uptake by subendothelial cells of the arterial wall and thereby decrease the risk of developing atherosclerotic plaque.

Our findings indicate that EX did not affect HDL-cholesterol concentration (*p* = 0.913). HDL turns over rather slow (half-life in the order of several days) [37], hence it is rather unlikely for any effects to manifest acutely, even though some studies have reported acute changes after exercise [38,39]. Chronic aerobic exercise training, however, can bring about a modest increase in HDL-cholesterol concentration [40]. The amount and intensity of exercise are important factors that mediate the beneficial effects of training on the concentration of HDL-cholesterol. Kraus et al. reported that HDL-cholesterol improved significantly only in the group of subjects who underwent a high amount of high-intensity exercise (8 months of jogging 32 km/week at 65–80% percent of peak oxygen consumption), but did not change significantly in the groups who underwent a low amount of high-intensity exercise (jogging 19.2 km/week at 65–80% of peak oxygen consumption) or a low amount of moderate-intensity exercise (walking 19.2 km/week at 40–55% of peak oxygen consumption) [31]. O’Donovan et al. also reported that there is no significant difference in improving HDL-cholesterol between high-intensity exercise (three 400-kcal sessions per week at 80% peak oxygen consumption) and moderate-intensity exercise (three 400-kcal sessions per week at 60% peak oxygen consumption), provided that the total energy expenditure was kept similar [41]. As our study was acute in nature, the absence of an effect of exercise on HDL-cholesterol concentration as well as HDL subclass distribution and particle size is not unexpected. On the other hand, we observed that CR caused a marginally significant decrease in HDL cholesterol (*p* = 0.049); however, none of the pairwise comparisons against the control trial reached significance. Our results regarding HDL-cholesterol concentration and HDL subclass number and distribution are consistent with those from previous studies [42,43,44]. 

We also did not observe any effects of acute energy deficits induced by diet or exercise on LDL particle concentration and subclass distribution. This is in agreement with a previous study on overweight and obese men and women, which found no changes in LDL subclass distribution and cholesterol concentration in large and small LDL particles 24 hours after a single bout of moderate-intensity exercise with an energy expenditure of 400 kcal [44]. Interestingly, another study in hypercholesterolemic men reported a mild ~6% increase in LDL-cholesterol concentration the day after a single session of exercise with an energy expenditure of 350 kcal [38]. Exercise, in our study, did appear to moderately raise the number of circulating LDL particles the next morning (Table 2), but none of these changes were statistically significant compared to the control trial. Our findings agree with the majority of studies on normocholesterolemic subjects, indicating that acute exercise does not affect LDL-C concentration the next day [38]. 

In clinical practice, the concentrations of LDL-cholesterol, HDL-cholesterol, and triglyceride are considered to be the primary hallmarks of dyslipidemia [3]. However, the plasma lipoprotein subclass distribution and particle size provide a further insight into CVD risk. For instance, a predominance of small, dense LDL particles is associated with an increased risk of CVD [13,45,46] as they penetrate the arterial subendothelial wall more easily [47] and are more susceptible to oxidation [48]. Our study showed that the negative energy balance induced by calorie restriction or exercise significantly decreased the number of circulating large and medium VLDL particles, with no changes in small VLDL particles. This resulted in a shift in the percent distribution of VLDL away from larger-sized particles, even though the average VLDL particle size did not change significantly. Although the clinical significance of this effect is not readily apparent, given the acute nature of our study, previous large-scale epidemiological studies have reported that a preponderance of larger VLDL particles is positively associated with CVD [16]. 

In conclusion, one day of acute energy deficit induced by CR or EX has a dose-dependent beneficial effect on blood lipid profile, which involves a linear reduction in the concentration of triglyceride in large and medium VLDL particles. The strengths of our study are the use of carefully controlled diet and exercise interventions, matched to the level of energy deficit. We found that acute diet- and exercise-induced energy deficits cause qualitatively and quantitatively similar changes in plasma lipid and lipoprotein profile. On the other hand, the acute nature of our interventions precludes making inferences about the long-term comparative therapeutic efficacy of diet and exercise. Furthermore, our study merely provides a static assessment of lipoprotein metabolism without an insight into the rates of *de novo* particle production, interconversion, or removal from circulation. Future studies should focus on understanding the mechanisms by which progressive calorie restriction and exercise reduce plasma triglyceride concentration, and whether combining the two interventions has additive beneficial effects.

## Figures and Tables

**Figure 1 nutrients-10-01814-f001:**
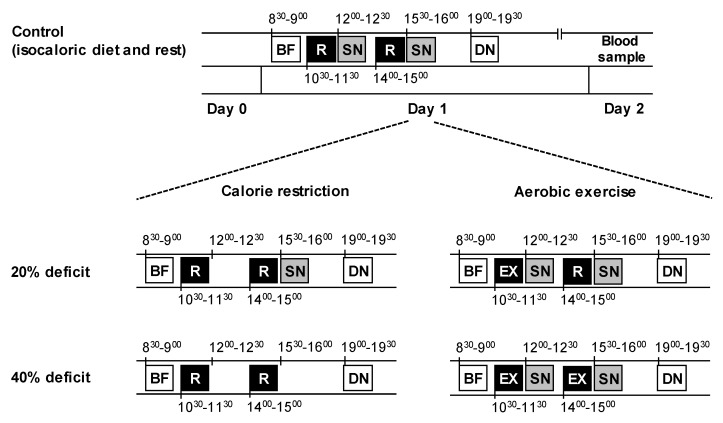
Experimental protocol. On day 1, subjects in both groups and all trials consumed an identical breakfast (BF) and dinner (DN), which provided 20% and 40% of the calories required for weight maintenance, respectively. Two snacks (SN), one in the morning and one in the afternoon, provided the remaining energy needed for weight maintenance (20% of calories each). In the diet group, subjects either consumed both snacks (control), one snack (20% deficit), or no snacks (40% deficit), and rested (R) in the morning and afternoon of day 1. In the exercise group, subjects either rested (control) or performed one bout (20% deficit) or two bouts (40% deficit) of aerobic exercise (EX), at approximately the same times of day as the corresponding resting periods in the diet group. After dinner, subjects fasted overnight and blood samples were obtained the next morning.

**Figure 2 nutrients-10-01814-f002:**
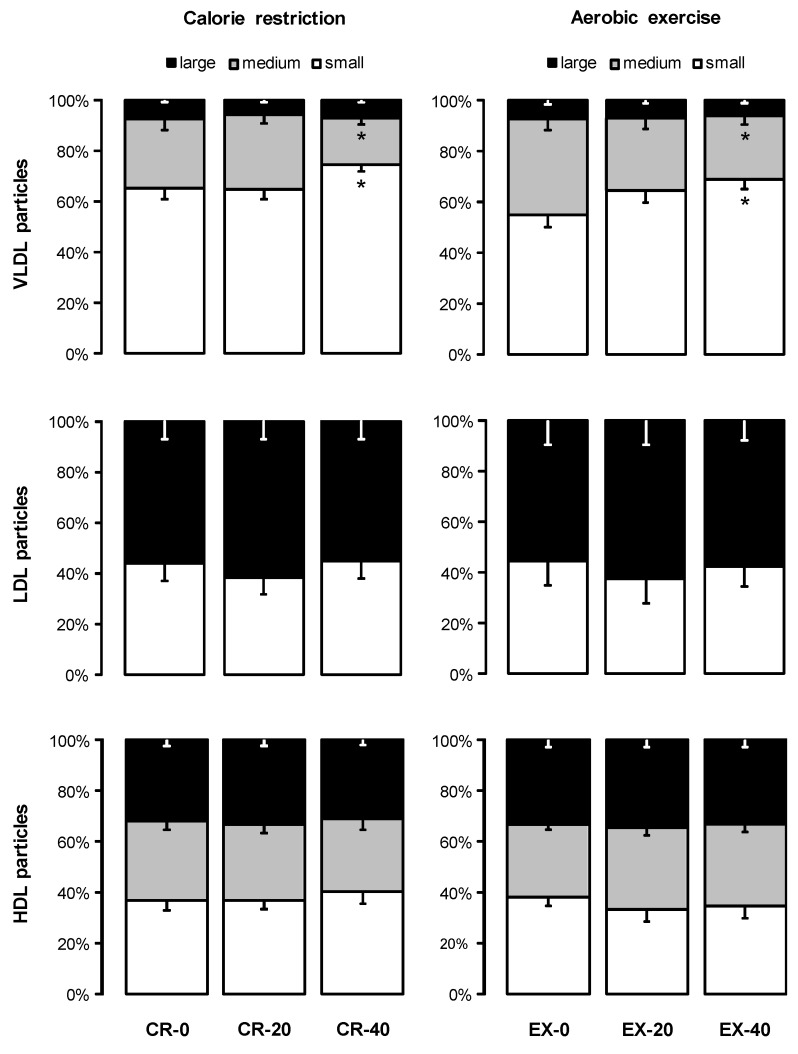
Effects of 20% and 40% energy deficits induced by calorie restriction (CR) and exercise (EX) on the relative distribution (% of total particles) of plasma lipoproteins. Data are mean ± SEM for *n* = 19 (CR group) and *n* = 13 (EX group). * The value is significantly different from the corresponding value in the control trial, *p* ≤ 0.05. Abbreviations: VLDL, very low-density lipoprotein; LDL, low-density lipoprotein; HDL, high-density lipoprotein.

**Table 1 nutrients-10-01814-t001:** Subject characteristics.

	Calorie Restriction (*n* = 19)	Exercise (*n* = 13)	*p*-Value
Sex (M/F)	7/12	3/10	0.41
Age (year)	26.4 ± 2.2	25.9 ± 2.6	0.89
Body mass index (kg/m^2^)	22.5 ± 1.7	20.9 ± 0.5	0.10
Body fat (%)	30 ± 1	32 ± 1	0.24
Fat mass (kg)	17.8 ± 1.1	17.7 ± 0.8	0.98
RMR (kcal)	1281 ± 68	1206 ± 47	0.38

Data are mean ± SEM; Abbreviations: M/F, male/female; RMR, resting metabolic rate.

**Table 2 nutrients-10-01814-t002:** Effects of diet- and exercise-induced energy deficits on lipoprotein concentrations, subclasses, and particle sizes.

	Calorie Restriction (*n* = 19)			Exercise (*n* = 13)		
	CR-0	CR-20	CR-40	EX-0	EX-20	EX-40
Total triglyceride (mg/dL)	76 ± 8	68 ± 6 *	63 ± 6 *	103 ± 13	84 ± 12 *	81 ± 10 *
VLDL-triglyceride (mg/dL)	63 ± 5	55 ± 4 *	52 ± 4 *	76 ± 8	62 ± 7 *	61 ± 6 *
HDL-cholesterol (mg/dL)	52 ± 3	53 ± 2	50 ± 2	56 ± 3	56 ± 3	55 ± 3
VLDL particles (nmol/L)	41.9 ± 3.0	40.2 ± 2.9	37.3 ± 2.6 *	49.0 ± 4.0	40.8 ± 3.6 *	43.4 ± 3.5 *
Large (nmol/L)	3.2 ± 0.4	2.3 ± 0.4 *	2.5 ± 0.3 *	3.7 ± 0.8	3.0 ± 0.7 *	2.7 ± 0.5 *
Medium (nmol/L)	12.4 ± 2.7	11.4 ± 1.4	7.1 ± 1.2 *	18.2 ± 2.3	12.2 ± 2.1 *	11.8 ± 2.1 *
Small (nmol/L)	26.3 ± 2.1	26.5 ± 2.6	27.7 ± 2.1	27.1 ± 3.4	25.6 ± 2.5	28.9 ± 1.8
IDL particles (nmol/L)	178 ± 18	176 ± 22	164 ± 22	248 ± 32	238 ± 22	225 ± 21
LDL particles (nmol/L)	529 ± 43	527 ± 47	551 ± 44	588 ± 64	622 ± 94	603 ± 82
Large (nmol/L)	293 ± 43	314 ± 39	289 ± 38	285 ± 42	327 ± 53	305 ± 28
Small (nmol/L)	236 ± 43	214 ± 41	263 ± 46	303 ± 83	294 ± 95	298 ± 81
HDL particles (μmol/L)	27.6 ± 1.1	27.5 ± 1.0	27.1 ± 0.9	29.2 ± 1.1	28.5 ± 1.2	28.6 ± 1.0
Large (μmol/L)	8.7 ± 0.7	9.1 ± 0.6	8.4 ± 0.6	9.6 ± 0.8	9.7 ± 0.9	9.4 ± 0.8
Medium (μmol/L)	8.3 ± 0.8	8.0 ± 0.8	7.5 ± 1.0	8.1 ± 0.5	8.9 ± 0.7	9.0 ± 0.7
Small (μmol/L)	10.7 ± 1.5	10.4 ± 1.1	11.2 ± 1.4	11.5 ± 1.4	9.8 ± 1.6	10.2 ± 1.6
VLDL size (nm)	49.2 ± 0.9	47.0 ± 1.2	46.7 ± 1.1	49.3 ± 1.3	47.7 ± 1.3	47.9 ± 1.0
LDL size (nm)	21.0 ± 0.1	21.1 ± 0.2	20.9 ± 0.1	21.1 ± 0.2	21.1 ± 0.2	21.0 ± 0.2
HDL size (nm)	9.9 ± 0.1	10.0 ± 0.1	9.9 ± 0.1	10.0 ± 0.1	10.0 ± 0.2	9.9 ± 0.1

Data are mean ± SEM. * The value is significantly different from the corresponding value in the control trial (CR-0 or EX-0), *p* ≤ 0.05. Abbreviations: CR, calorie restriction; EX, exercise; VLDL, very low-density lipoprotein; IDL, intermediate-density lipoprotein; LDL, low-density lipoprotein; HDL, high-density lipoprotein.

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
