# Peer review of "Lipoprotein Subclass Profile after Progressive Energy Deficits Induced by Calorie Restriction or Exercise"

_nutrients, 2018, doi:10.3390/nu10111814_

Round 1
Reviewer 1 Report
General Comments
This was a very interesting study to read about, with an interesting study design, to compare calorie restriction and exercise in a dose-dependent fashion in a good number of subjects. However, there are several areas in which the manuscript requires improvement. There is no discussion of the impact of the large measurement error noted (high CVs) for some measurements, mainly VLDL. The washout period was variable and may not have been sufficient to avoid any potential carryover effects. The exact timing of snacks is not detailed. Also, further discussion of the potential mechanisms by which calorie restriction and exercise may have elicited the observed effects is needed, even if speculative.
Introduction
(I have had to use quotations here as the manuscript did not have line numbers.)
- "Lifestyle factors such as...key in reducing CVD risk" - Please supple reference(s)
- "...by a chronic state of negative energy balance, involves tipping the balance between energy intake and energy expenditure" - in which direction?
- "Clearly, however, diet and exercise can affect lipid metabolism even in the absence of weight loss (9, 10)" - Although citations are provided, please clarify how diet and exercise effects are independent of weight loss in one/two sentences, or a bit more detail.
- Little known about comparative effects of weight loss and energy deficit - this was investigated in the study of Maraki et al (2010), yet it is not specifically mentioned. Suggest brief mention of their findings and how the current study builds on this.
Methods
Subjects
- What was the mean (SE) BMI of the participants? Suggest including in a table of baseline anthropometrics in results section.
- Why were there uneven numbers in the two study arms?
- Were the individuals matched on the basis of age, BMI, percent body fat etc.?
- What was the ethnicity/race of the subjects?
Experimental design
- How were the initial resting metabolic rate and body composition measured?
- Why was the period between study visits so variable (5-10 days)? Was this sufficient as to avoid any potential 'carryover' effects between visits?
- 'Body composition and resting energy expenditure' - should be a sub-heading.
Blood sampling
- There are not three different trials - there are different phases, visits or treatments of the same trial, which has two parallel arms.
- Subjects replicated the same diet as before their first visit - how was this assessed or monitored to ensure they consumed the same foods/energy?
- Between study day and the following morning's blood sample, did subjects again refrain from exercise, alcohol and caffeine?
- Figure 1 - what was the exact timing of the snacks and was this maintained between visits? Figure doesn't indicate that the same protocol was repeated for the other study visits
- When were the snacks administered?
-What foods were used to compose the test diets?
- This design allowed for inducing progressively increasing negative energy balance (equal to 20% and 40%) by restricting dietary intake (CR group) or by increasing energy expenditure (EX group), at approximately the same time of the day - Why approximately the same time of day? This should have been done at precise the same time during each visit.
Sample analyses
- It is assumed that particles with a size >60nm are large VLDL. In fact, this would have included some chylomicron particles, even in the fasted state in healthy individuals. This should be mentioned here as an assumption.
- What are the CV values for reproducibility given here? Are these within- or between-subject values?
- The CV values for Large (14%) and medium (21%) sized VLDL particles are very high. If the within-subject reproducibility is so poor, can any differences between study visits/treatments really be inferred?
- Can NMR really achieve the degree of sensitivity implied by the units for VLDL, IDL and LDL particles (Table 1)? These values are of the order of picomoles per litre, with differences between means of<10 picomoles/L.
- More detail required on how plasma and VLDL-triglycerides and HDL-cholesterol are measured via NMR, if only briefly. What is the 'conventional' method for each of these? If this is via an enzymatic autoanalyser method, isn't this cheaper and more widely used in current clinical practice?
Statistical Analysis
- Clarify why Shapiro-Wilks test was performed and its impact on the decision to use either parametric or non-parametric analyses.
- Were non-normally distributed variables transformed prior to analysis?
- More detail on the logic of the statistical approach taken is required, including which variables were analysed using which statistical tests.
- Test results for differences between groups should be included in the results section in a table of subjects' baseline characteristics.
Results
In general, a table of the subjects' baseline anthropometrics is required.
Energy deficits induced by CR and EX
- Inconsistent significant figures between energy balance values quoted for CR and EX groups.
Lipoprotein subclass profile
- The differences between groups are very small relative to the reported measurement error. Although it is mentioned that VLDL and IDL are typically low in (fasted?) plasma, this is not a justification for the high CV values reported. If you cannot measure the parameter with sufficient accuracy it should not be measured!
The measurement error and its impact on the results and their interpretation requires further justification and comparison with previous similar measurements.
Discussion
- "...but diet and exercise have never been directly compared" - Yes, they have in the study of Maraki et al (2010), which is cited here as reference number 10!
- Shift in the percent distribution of VLDL particles from larger to smaller VLDL without a change in the average (mean?) particle size - Is this therefore of interest?
- There is no discussion of the potential impact of the large reported measurement error on the results and their interpretation.
- A shift towards a greater number of smaller VLDL particles may only be desirable if they are effectively cleared from the circulation and do not form IDL and VLDL particles. Some more in depth discussion of the potential effects of CR and EX on particle metabolism, including TAG hydrolysis by lipoprotein lipase or hepatic lipase, and particle production and clearance, is required.
- Some discussion and/or speculation on potential mechanisms for the shift in particle size is required. For example, is this due to a greater uptake of triglycerides by heart and skeletal muscle during exercise? Why are there no differences in the effects of exercise and calorie restriction and how may they differentially exert these effects?
Author Response Reviewer 1:
General Comments
This was a very interesting study to read about, with an interesting study design, to compare calorie restriction and exercise in a dose-dependent fashion in a good number of subjects. However, there are several areas in which the manuscript requires improvement. There is no discussion of the impact of the large measurement error noted (high CVs) for some measurements, mainly VLDL. The washout period was variable and may not have been sufficient to avoid any potential carryover effects. The exact timing of snacks is not detailed. Also, further discussion of the potential mechanisms by which calorie restriction and exercise may have elicited the observed effects is needed, even if speculative.
Introduction
(I have had to use quotations here as the manuscript did not have line numbers.)
- "Lifestyle factors such as...key in reducing CVD risk" - Please supple reference(s)
Reply: We have added citations to this statement. (page 2)
- "...by a chronic state of negative energy balance, involves tipping the balance between energy intake and energy expenditure" - in which direction?
Reply: We have revised this statement to make it clearer. (page 2)
- "Clearly, however, diet and exercise can affect lipid metabolism even in the absence of weight loss (9, 10)" - Although citations are provided, please clarify how diet and exercise effects are independent of weight loss in one/two sentences, or a bit more detail.
Reply: We have revised this statement to make it clearer. (page 2)
- Little known about comparative effects of weight loss and energy deficit - this was investigated in the study of Maraki et al (2010), yet it is not specifically mentioned. Suggest brief mention of their findings and how the current study builds on this.
Reply: The original paper by Maraki (2010) is a review paper and not an original investigation. We are not aware of any paper that specifically evaluated the effects of weight loss vs energy deficit on lipid profile. After reading a later comment of yours, we realized you refer to a different paper by the same author which we cited later on in the manuscript.
Methods
Subjects
- What was the mean (SE) BMI of the participants? Suggest including in a table of baseline anthropometrics in results section.
Reply: We have included a table with baseline characteristics of the participants. (Table 1 on page 6)
- Why were there uneven numbers in the two study arms?
Reply: Initially, 15 subjects were recruited in each group, at which point we performed an interim analysis and decided to recruit an additional 5 subjects in the diet group, because the effect size turned out to be smaller than the one we assumed to adequately power the study. As a result, a total of 15 subjects were recruited in the exercise group (2 dropped out and did not complete all trials, hence 13 completers were analyzed) and a total of 20 subjects were recruited in the diet group (1 dropped out and did not complete all trials, hence 19 completers were analyzed). We have now added this information in the Methods of the revised manuscript. (page 2)
- Were the individuals matched on the basis of age, BMI, percent body fat etc.?
Reply: The two groups were not specifically matched for these attributes a priori, but we did not detect any differences in their baseline characteristics, such as age (P=0.89), sex (P=0.41), BMI (P=0.10), percent body fat (P=0.24), fat mass (P=0.98), and RMR (P=0.38). We now mention this in the Results section of the revised manuscript (text and Table 1 on page 6).
- What was the ethnicity/race of the subjects?
Reply: Most (30 out of 32) subjects were Singaporeans of Chinese descent; the other two included one Vietnamese and one Indian. We have included this information in the revised manuscript. (page 5)
Experimental design
- How were the initial resting metabolic rate and body composition measured?
Reply: We have included information about the measurement of RMR and body composition in the Methods of the revised manuscript. (page 3)
- Why was the period between study visits so variable (5-10 days)? Was this sufficient as to avoid any potential 'carryover' effects between visits?
Reply: The washout period was chosen as a compromise between subject availability, scheduling openings in the research center, and absence of any evidence suggesting that a single day of exercise or calorie restriction can have such delayed effects (beyond 5 days). Furthermore, since trials were performed in random order, any carry-over effects (if any) would not lead to a systematic difference among trials.
- 'Body composition and resting energy expenditure' - should be a sub-heading.
Reply: We have revised this accordingly. (page 3)
Blood sampling
- There are not three different trials - there are different phases, visits or treatments of the same trial, which has two parallel arms.
Reply: We have replaced “trials” for “visits” in the revised manuscript. (page 3)
- Subjects replicated the same diet as before their first visit - how was this assessed or monitored to ensure they consumed the same foods/energy?
Reply: Upon their first visit, subjects provided a food diary, which was given back to them as a diet prescription when they were discharged. They were instructed to eat the same food for their subsequent visits. All subjects confirmed they did so. We have added this detail in the revised manuscript. (page 3)
- Between study day and the following morning's blood sample, did subjects again refrain from exercise, alcohol and caffeine?
Reply: Subjects were discharged late on the study day, after eating dinner, and were instructed to refrain from all food and drinks, as well as physical activity until they were readmitted the next morning. We have added this information in the revised manuscript. (page 3)
- Figure 1 - what was the exact timing of the snacks and was this maintained between visits? Figure doesn't indicate that the same protocol was repeated for the other study visits
Reply: We have redrawn Figure 1 to make it clear. (page 4)
- When were the snacks administered?
Reply: Snacks were administered at 12:00pm and 3.30pm (shown in revised Figure 1, page 4).
-What foods were used to compose the test diets?
Reply: Breakfast: turkey breast sandwich with soya milk; snacks: tuna sandwich with orange juice; dinner: teriyaki chicken rice with mixed vegetables and peaches. Each meal or snack contained 55% of total energy as carbohydrate, 27% as fat, and 18% as protein. We have added this information in the revised manuscript. (pages 3 and 4)
- This design allowed for inducing progressively increasing negative energy balance (equal to 20% and 40%) by restricting dietary intake (CR group) or by increasing energy expenditure (EX group), at approximately the same time of the day - Why approximately the same time of day? This should have been done at precise the same time during each visit.
Reply: The diet- and exercise-induced energy deficits could not have occurred at the same times of day, because snacks were consumed at exactly the same time in both groups, and the addition of exercise in the exercise group could not coincide with the time of snacks (refer to revised Figure 1 on page 4).
Sample analyses
- It is assumed that particles with a size >60nm are large VLDL. In fact, this would have included some chylomicron particles, even in the fasted state in healthy individuals. This should be mentioned here as an assumption.
Reply: Indeed the fasting samples may contain some residual chylomicrons. We have added this detail in the revised manuscript. (page 5)
- What are the CV values for reproducibility given here? Are these within- or between-subject values?
Reply: These are CVs of repeated measurements on the same samples (i.e. within). We have added this detail in the revised manuscript. (page 5)
- The CV values for Large (14%) and medium (21%) sized VLDL particles are very high. If the within-subject reproducibility is so poor, can any differences between study visits/treatments really be inferred?
Reply: The relatively high CVs are due to the fact these subclasses (e.g. large VLDL) are present in very small concentrations in fasting plasma (refer to Table 2). Please note that we detected significant differences between trials in both diet and exercise groups for large VLDL. This implies that the effects are detectable despite the relatively large CV, meaning they are robust and reproducible.
- Can NMR really achieve the degree of sensitivity implied by the units for VLDL, IDL and LDL particles (Table 1)? These values are of the order of picomoles per litre, with differences between means of<10 picomoles/L.
Reply: We are certain the reviewer acknowledges that our study is not the first to utilize NMR to measure lipoprotein subclasses in plasma. Measurements were performed at LipoScience (owned by LabCorp) which is the spin-off company of James Otvos, who pioneered this technique and keeps improving the relevant technology. We have no reason to question the sensitivity of NMR determinations.
- More detail required on how plasma and VLDL-triglycerides and HDL-cholesterol are measured via NMR, if only briefly. What is the 'conventional' method for each of these? If this is via an enzymatic autoanalyser method, isn't this cheaper and more widely used in current clinical practice?
Reply: The only conventional method we can think of for analysis of lipoprotein subclasses in plasma would require utilizing sequential steps of ultracentrifugation to separate the different subclasses, which is technically far more challenging, and thus more variable and prone to measurement errors. These methods are not being used in clinical practice.
Statistical Analysis
- Clarify why Shapiro-Wilks test was performed and its impact on the decision to use either parametric or non-parametric analyses.
Reply: We have revised the statistical analysis section to clarify this. (page 5)
- Were non-normally distributed variables transformed prior to analysis?
Reply: We did not employ parametric tests on transformed data. Instead, we used non-parametric tests for non-normally distributed data (these tests rely on ranking the data). We revised the manuscript to clarify this. (page 5)
- More detail on the logic of the statistical approach taken is required, including which variables were analysed using which statistical tests.
Reply: As we mention in the manuscript, the primary goal of our study was to determine the effects of progressively increasing energy deficit induced by calorie restriction or aerobic exercise on plasma lipoproteins. This was accomplished by using analysis of variance for repeated measures within each study group. Statistically significant models were followed by i) trend analysis to describe the pattern of change in the outcome with progressively increasing negative energy balance (i.e., linear or quadratic), and ii) simple contrasts to compare each level of energy deficit against the control trial. We mention these details in the statistical analysis section. (page 5)
- Test results for differences between groups should be included in the results section in a table of subjects' baseline characteristics.
Reply: We have conducted this additional analysis, as suggested, and found no significant differences between groups in baseline lipid and lipoprotein concentrations and lipoprotein subclass profile. We have added this detail in the revised manuscript. (page 6)
Results
In general, a table of the subjects' baseline anthropometrics is required.
Reply: We have included such a table in the revised manuscript (Table 1 on page 6).
Energy deficits induced by CR and EX
- Inconsistent significant figures between energy balance values quoted for CR and EX groups.
Reply: The diet induced energy deficits were precisely 20% and 40% in all subjects (no variation), as they only depended on the diet calculations. The corresponding figures shown for exercise are based on the actual calories expended, which were computed after the exercise was done (which is why there is some variation from the theoretically computed 20% and 40% deficits). This is why we opted to show one decimal for exercise-induced negative energy balance and no decimals for diet-induced negative energy balance.
Lipoprotein subclass profile
- The differences between groups are very small relative to the reported measurement error. Although it is mentioned that VLDL and IDL are typically low in (fasted?) plasma, this is not a justification for the high CV values reported. If you cannot measure the parameter with sufficient accuracy it should not be measured!
Reply: As we mentioned above, the validity of NMR has been demonstrated by numerous previous studies, and we have no reason to question the methodology. A high CV, as for large VLDL, will make it more difficult to detect significant differences. However, we detected significant differences for large VLDL in both energy deficit trials and in both groups. This is a testament that the effects are both true and robust.
The measurement error and its impact on the results and their interpretation requires further justification and comparison with previous similar measurements.
Reply: We are not aware of any equivalent methodologies that can measure all lipoprotein subclasses. Other methods can only measure limited outcomes, such as LDL particle concentration by immunoturbidimetry after ultracentrifugation to remove VLDL, or LDL and HDL sizes by gradient gel electrophoresis after sequential ultracentrifugation. These methods perform similarly to NMR and the two measurements always correlate well with each other. Much information about the validation of NMR and its comparison to other methods has been published previously (refs 1-3 below; also cited in our revised manuscript). We do not anticipate the measurement error to have had an impact on our results, since we detected significant differences in the outcomes with larger measurement error, whereas we found no differences in outcomes with smaller measurement error.
Matyus SP, Braun PJ, Wolak-Dinsmore J, et al. NMR measurement of LDL particle number using the Vantera Clinical Analyzer. Clin Biochem 2014;47:203-10.
Matyus SP, Braun PJ, Wolak-Dinsmore J, et al. HDL particle number measured on the Vantera(R), the first clinical NMR analyzer. Clin Biochem 2015;48:148-55.
Jeyarajah EJ, Cromwell WC, Otvos JD. Lipoprotein particle analysis by nuclear magnetic resonance spectroscopy. Clin Lab Med 2006;26:847-70.
Discussion
- "...but diet and exercise have never been directly compared" - Yes, they have in the study of Maraki et al (2010), which is cited here as reference number 10!
Reply: Original ref. 10 (now ref 12) was a review paper. We presume the reviewer refers to the study by Maraki in Clinical Nutrition 2010; 29: 459-463. Indeed this is correct. We revised our manuscript as follows: “…but possible dose-response relationships between diet- and exercise-induced energy deficits have never been investigated.” We also now cite the above paper, which agrees with our findings. (page 9)
- Shift in the percent distribution of VLDL particles from larger to smaller VLDL without a change in the average (mean?) particle size - Is this therefore of interest?
Reply: From the physiological perspective, we believe it is. From the clinical perspective, average particle size is not the only parameter of interest. In fact, studies have shown that a preponderance of large VLDL particles (i.e. more large VLDL particles without necessarily altered average particle size) is associated with increased risk for cardiovascular disease, for example:
Freedman DS, Otvos JD, Jeyarajah EJ, Barboriak JJ, Anderson AJ, Walker JA. Relation of lipoprotein subclasses as measured by proton nuclear magnetic resonance spectroscopy to coronary artery disease. Arterioscler Thromb Vasc Biol 1998;18:1046-53.
We comment about this issue in the Discussion of the revised manuscript. (page 10)
- There is no discussion of the potential impact of the large reported measurement error on the results and their interpretation.
Reply: Please refer to our detailed response to an earlier comment. We do not anticipate the measurement error to have had an impact on our results, since we detected significant differences in the outcomes with larger measurement error (i.e. large and medium VLDL), whereas we found no differences in outcomes with smaller measurement error.
- A shift towards a greater number of smaller VLDL particles may only be desirable if they are effectively cleared from the circulation and do not form IDL and VLDL particles. Some more in depth discussion of the potential effects of CR and EX on particle metabolism, including TAG hydrolysis by lipoprotein lipase or hepatic lipase, and particle production and clearance, is required.
Reply: Our data suggest a shift towards smaller VLDL in relative terms (% of total) and not a shift towards a greater number of smaller VLDL (small VLDL particle concentrations were the same in all trials). Given we did not detect any changes in IDL and LDL concentrations, it could be inferred that there was no increased formation of IDL and LDL. Our measurements of particle concentrations, however, only provide a static measure of lipoprotein metabolism. We did not employ methodologies (e.g. isotope tracers) to evaluate flux rates between different subclasses of the same lipoprotein or between different lipoproteins. We have commented on this limitation in the revised manuscript. (page 10)
- Some discussion and/or speculation on potential mechanisms for the shift in particle size is required. For example, is this due to a greater uptake of triglycerides by heart and skeletal muscle during exercise? Why are there no differences in the effects of exercise and calorie restriction and how may they differentially exert these effects?
Reply: We have added a short description of the mechanisms by which acute diet and exercise could affect the lipoprotein fractions of interest, i.e. VLDL particles and triglyceride. (page 9)

Reviewer 2 Report
The effect of acute energy restriction on lipoprotein subclass was not established yet. It is interesting to investigate it. However, there are several considered problems on the study design as follows.
Major comments:
1) Some researchers have demonstrated the similar work for long-term period. Even if diet restriction can be done at a day, what are the clinical meanings of the change of lipoprotein subclass? The reviewer does not understand the meanings and experimental design because people usually continue the restriction diet for at least several days, weeks or months. Other publications exhibited that VLDL particles were unchanged under low carbohydrate diet for 3-months (Rodriguez-Garcia et al. Medicine, 96:27, 2017). On the other research, VLDL particles were increased under low carbohydrate or low fat diet for 12-months compared to 3-months (LeCheminant et al. Lipids in Health and Disease, 9:54, 2010). These reports could show that lipoprotein profiles in restriction diet for long term was different from those for short term like the author did in this experiment. Also, regarding changes of VLDL particles, the author did not discuss enough the difference among previous reports and present ones.
2) The author mentioned that diet and exercise have never been directly compared (Line 4, Discussion). Mechanism of biological changes derived from energy restriction was surely different from the changes derived from exercise. How do you control/compare the condition between calorie restriction and exercise? It should be difficult to compare two approaches.
3) The levels of HDL particle were 1,000 times higher than those of VLDL particles, probably because of serum from healthy subjects. Extend of change of VLDL particles seems within error.
4) Discussion on unchanged HDL particles that the author did discuss was too much volume. However, it is not enough to discuss metabolically on other lipoprotein subclass such as tendency of increases of small LDL in calorie restriction.
5) The author should discuss on the possible mechanism of decreased large and medium VLDL particles from the point of lipoprotein lipase (LPL) and hepatic lipase, which is responsible for hydrolysis of triglycerides in plasma lipoproteins and a process generating fatty acids for storage or energy production. For example, exercise induces lipoprotein lipase.
6) Half-life in plasma should be 5 days in HDL and a few days in LDL. The author should have taken at least 5days for wash-out. Diet just before beginning may affect lipoprotein profile in such an acute diet test.
Minor comments:
1) It was hard to read the time schedule, grouping, fasting period in Figure 1. Please improve it.
Author Response Reviewer 2:
The effect of acute energy restriction on lipoprotein subclass was not established yet. It is interesting to investigate it. However, there are several considered problems on the study design as follows.
Major comments:
1) Some researchers have demonstrated the similar work for long-term period. Even if diet restriction can be done at a day, what are the clinical meanings of the change of lipoprotein subclass? The reviewer does not understand the meanings and experimental design because people usually continue the restriction diet for at least several days, weeks or months. Other publications exhibited that VLDL particles were unchanged under low carbohydrate diet for 3-months (Rodriguez-Garcia et al. Medicine, 96:27, 2017). On the other research, VLDL particles were increased under low carbohydrate or low fat diet for 12-months compared to 3-months (LeCheminant et al. Lipids in Health and Disease, 9:54, 2010). These reports could show that lipoprotein profiles in restriction diet for long term was different from those for short term like the author did in this experiment. Also, regarding changes of VLDL particles, the author did not discuss enough the difference among previous reports and present ones.
Reply: We certainly appreciate that our acute calorie restriction intervention cannot be used to make inferences about long-term calorie restriction (which typically leads to weight loss). In addition, as the reviewer points out, the composition of the diet (low fat or low carbohydrate) can also affect VLDL metabolism independent of the amount of calorie deficit per se. In our study, we only manipulated the amount of calories while maintaining the diet composition the same across all trials. So our study design is different both “quantitatively” (acute vs chronic diet) and “qualitatively” (same or different macronutrient composition) from the studies mentioned above. Long term diets with different composition may well lead to different responses of VLDL metabolism than our acute calorie restriction, however there are many other factors involved in a long-term diet study that make comparisons difficult, particularly as changes in body weight, body composition and body fat distribution. Although we do not question the validity of long-term diet trials, an expanded discussion about the effects of diet on lipoprotein metabolism falls beyond the scope of our paper, which was to compare acute energy deficits induced by diet and exercise. Nevertheless, we revised our manuscript throughout to discuss more the results from previous acute studies, and highlight the fact our results may not necessarily be applicable in the long term (pages 9 and 10).
2) The author mentioned that diet and exercise have never been directly compared (Line 4, Discussion). Mechanism of biological changes derived from energy restriction was surely different from the changes derived from exercise. How do you control/compare the condition between calorie restriction and exercise? It should be difficult to compare two approaches.
Reply: This was exactly the purpose of our study. To compare the effects of the same energy deficits induced by diet and exercise on lipoprotein subclass concentration, distribution and size. The reviewer is correct that even if the changes induced by diet and exercise were similar (i.e. reduction of VLDL triglyceride concentration associated with fewer circulating large VLDL particles), the biological mechanisms may be different for diet and exercise. In fact, the mechanism for the exercise-induced reduction in VLDL particle number and VLDL triglyceride concentrations likely involve a decrease in the production of VLDL particles from the liver and an increase in the clearance of VLDL triglyceride from the periphery. On the contrary, chronic calorie restriction (leading to weight loss) lowers plasma triglyceride and VLDL triglyceride concentrations predominantly by reducing hepatic VLDL triglyceride secretion. Unfortunately, we did not assess the underlying mechanisms for the changes we observed in lipid and lipoprotein profile in our study. We have revised our Discussion to mention these issues (page 9).
3) The levels of HDL particle were 1,000 times higher than those of VLDL particles, probably because of serum from healthy subjects. Extend of change of VLDL particles seems within error.
Reply: The number of circulating HDL particles in indeed much greater than the number of VLDL particles. The reduction in the number of large VLDL particles may be small within the context of how many lipoprotein particles circulate in plasma, however this represented a 20-30% change. Importantly, this effect was consistently found in both energy deficit trials (vs control) and in both groups (diet and exercise), and was in agreement with the corresponding reductions in plasma triglyceride concentrations (since most triglyceride in plasma is carrier in VLDL). We are therefore confident these effects are robust.
4) Discussion on unchanged HDL particles that the author did discuss was too much volume. However, it is not enough to discuss metabolically on other lipoprotein subclass such as tendency of increases of small LDL in calorie restriction.
Reply: We have focused our discussion on VLDL and HDL simply because VLDL-triglyceride and HDL-cholesterol are the lipid fractions predominantly affected by exercise. The reviewer is correct that in doing so, we neglected LDL. We have therefore revised our manuscript to discuss LDL as well. (page 10).
5) The author should discuss on the possible mechanism of decreased large and medium VLDL particles from the point of lipoprotein lipase (LPL) and hepatic lipase, which is responsible for hydrolysis of triglycerides in plasma lipoproteins and a process generating fatty acids for storage or energy production. For example, exercise induces lipoprotein lipase.
Reply: We have revised our manuscript to discuss possible mechanisms as suggested. (page 9).
6) Half-life in plasma should be 5 days in HDL and a few days in LDL. The author should have taken at least 5days for wash-out. Diet just before beginning may affect lipoprotein profile in such an acute diet test.
Reply: Indeed, the reviewer is correct. This is why we chose a washout period between 5 and 10 days (6-11 days between the actual sampling days). Regarding the diet before the study, subjects provided a food diary upon their first visit, which was given back to them as a diet prescription when they were discharged. They were instructed to eat the same food before they come in for their subsequent visits. All subjects confirmed they did so. We have added theses details in the revised manuscript. (page 3)
Minor comments:
1) It was hard to read the time schedule, grouping, fasting period in Figure 1. Please improve it.
Reply: We have redrawn Figure 1 to make it clearer. (page 4)

Reviewer 3 Report
The author determined the effects of acute energy deficits by calorie restriction or endurance exercise on blood lipoprotein profiles. Obtained data indicate that acute negative energy balance reduce triglyceride concentrations in a dose-dependent manner, by decreasing circulating large and medium VLDL particles.
Major concern
1) Please resend page 7 of the following manuscript because the right quarter of the page is displayed outside the page and can not be read the result section.
2) Please indicate the physiological characteristics (age, sex, body weight, BMI, fat mass, fat free mass) of subjects in calorie restriction group and exercise group, separatedly.
Author Response Reviewer 3:
The author determined the effects of acute energy deficits by calorie restriction or endurance exercise on blood lipoprotein profiles. Obtained data indicate that acute negative energy balance reduce triglyceride concentrations in a dose-dependent manner, by decreasing circulating large and medium VLDL particles.
Major concern
1) Please resend page 7 of the following manuscript because the right quarter of the page is displayed outside the page and can not be read the result section.
Reply: We apologize for this. The journal editors copyedited our manuscript and page 7 was in landscape format. We hope it looks ok now.
2) Please indicate the physiological characteristics (age, sex, body weight, BMI, fat mass, fat free mass) of subjects in calorie restriction group and exercise group, separatedly.
Reply: We have included a table with baseline characteristics of the participants. (Table 1 on page 6). There were no significant differences between groups.

Round 2
Reviewer 1 Report
I would like to thank the authors for the significant improvements made to the manuscript; in particular with regards to the explanation of the clinical protocol and the discussion of the findings. The authors have provided very detailed responses to all queries raised in Report 1. I am now satisfied that the paper should be advanced.
I would like to congratulate the authors on conducting a valuable study in an area which has a very immediate and important clinical and public health relevance.
Author Response Reviewer 1:
Thank you for your kind words.
Reviewer 2 Report
No concerns for publication. the authors corrected the manuscript adequately.
Reviewer 3 Report
The description of the research design became very clear.
1) Energy restriction by meal and energy consumption by exercise are equal. On the other hand, there is a lack of changes in the intake of three major nutrients due to dietary restriction and the description of carbohydrate and lipid consumption by exercise. Therefore, it is unclear whether the conclusions obtained this time can be explained only by changes in energy intake / consumption.
2) In addition, it is necessary to describe the PFC ratio in 1688 ± 66 kcal / day of ordinary diet.
Author Response Reviewer 3:
1) Energy restriction by meal and energy consumption by exercise are equal. On the other hand, there is a lack of changes in the intake of three major nutrients due to dietary restriction and the description of carbohydrate and lipid consumption by exercise. Therefore, it is unclear whether the conclusions obtained this time can be explained only by changes in energy intake / consumption.
Reply: We acknowledge this concern. Typically, exercise at moderate intensity in untrained subjects in accompanied by Respiratory Exchange Ratios of ~0.9 [Bergman (1999) J Appl Physiol 86(2)479], which means a contribution to total energy expenditure of 33% fat and 67% carbohydrate. The diet-induced energy deficit was the same composition as the diet, i.e. 55% of total energy as carbohydrate, 27% as fat, and 18% as protein. Matching the energy deficits for macronutrient composition means we would have to provide a different diet in the exercise trials, which would add another variable in the interpretation of our findings. We feel this small difference in the macronutrient composition of the diet vs exercise deficits cannot have confounded our findings. Nevertheless, we added a comment on this limitation. (page 6 of the re-revised manuscript)
2) In addition, it is necessary to describe the PFC ratio in 1688 ± 66 kcal / day of ordinary diet.
Reply: We mention the PCF ratio of the diet on pages 4-5 of the re-revised manuscript.